# AMSD: The Australian Message Stick Database

**Piers Kelly** [1,2]*, **Junran Lei**[3], **Hans-Jörg Bibiko**[4], **Lorina Barker**[5]

1 Department of Archaeology, Classics and History, University of New England, Armidale, New South Wales, Australia, 2 Centre for Australian Studies, Universität zu Köln, Cologne, North Rhine-Westphalia, Germany, 3 Centre for Digital Humanities Research, Australian National University, Canberra, Australian Capital Territory, Australia, 4 Department of Linguistic and Cultural Evolution, Max Planck Institute for Evolutionary Anthropology, Leipzig, Saxony, Germany, 5 School of Humanities, Arts, and Social Sciences, University of New England, Armidale, New South Wales, Australia

* pkelly26@une.edu.au

**Data Availability Statement:** Data is available publicly here: https://amsd.clld.org/.

**Funding:** The lead author (Piers Kelly) receives salary and project funding specifically for the research described in this paper. He is funded by an ARC Discovery Early Career Researcher Award

## Abstract

Message sticks are wooden objects once widely used in Indigenous Australia for facilitating important long-distance communications. Within this tradition an individual wishing to send a message would carve a stick and apply conventional symbols to its surface. The stick was entrusted to a messenger who carried the object into the territory of another community together with a memorised oral statement. Between the 1880s and the 1910s, settlers and international scholars took great interest in message sticks and this was reflected in efforts to document, collect and store them in museums worldwide. However, by this period, the practice was already undergoing profound changes, having been abandoned in many parts of the continent and transformed in others. While message sticks were still being used in a traditional way in Western Arnhem Land up until at least the late 1970s, today they feature in public interactions between Indigenous and non-Indigenous organisations, in art production and in oral narrations. Accordingly many questions concerning the history, pragmatics and global significance of message stick communication remain unanswered. To address this we have compiled the Australian Message Stick Database, a new resource hosted at the Max Planck Institute for Evolutionary Anthropology, Leipzig, and The Australian National University, Canberra. It contains images and data for over 1500 individual message sticks sourced from museums, and supplemented with information derived from published and unpublished manuscripts, private collections, and from field recordings involving contemporary Indigenous consultants. For the first time, knowledge about Australian message sticks can be evaluated as a single set allowing scholars and Traditional Owners to explore previously intractable questions about their histories, meanings and purposes.

## Introduction

Australian message sticks are public communication devices once used across almost all regions of Indigenous Australia, and now held in over 20 museums and private collections worldwide. The Australian Message Stick Database (https://amsd.clld.org/; hereafter the AMSD), compiled by Indigenous and non-Indigenous scholars, is a new resource that aims to

with the grant number DE220100795. No other author has received specific funding for this work. The funders had no role in study design, data collection and analysis, decision to publish, or preparation of the manuscript.

**Competing interests:** No authors have competing interests

unite all known surviving artefacts in collecting institutions together with archival materials that explain their uses and meanings. Applying standardised metadata, the AMSD can be consulted to reveal information on distribution, lexical terminology, makers, collection histories, motifs, materials, meanings and sources. In aggregate, this information will allow researchers to investigate message stick communication as a coherent system of long-distance communication that connected Australia's First Nations across geographical, cultural and linguistic space.

Message sticks are hand-made wooden objects that are generally either flattened or cylindrical, and sometimes tapered at both ends (Fig 1; for a review see [1]). Designed as communication aids, they are marked with meaningful motifs that are engraved or painted, or a combination of both. Within these broad descriptive parameters they are remarkably diverse in terms of their formal characteristics. The measured objects (N = 626) in the Australian Message Stick Database range in size from 35mm to 1860mm, with a median length of 170mm. All but two are made of wood (the exceptions are bone and clay), and a small number are

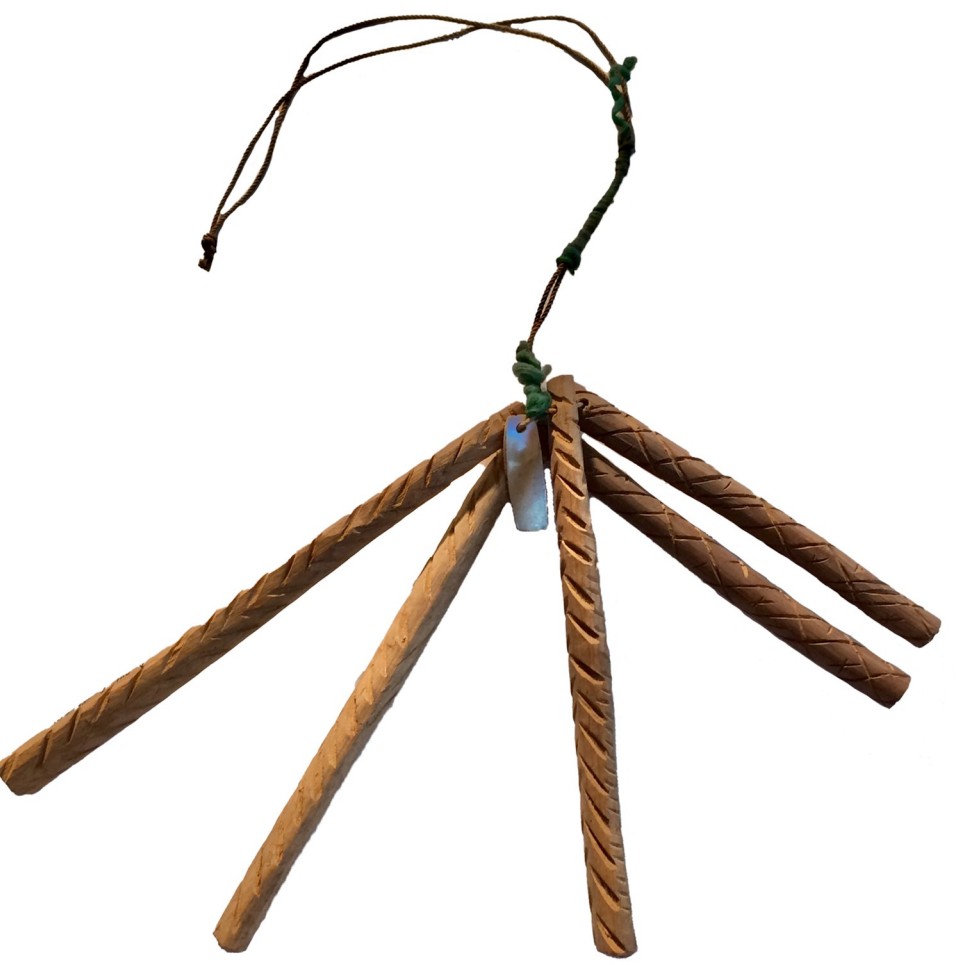

**Fig 1. A set of five message sticks linked with string, held at the Grassi Museum Leipzig (AMSD ID: GML_Au941).** Provenance is not secure, but the set was plausibly acquired by Herman Klaatsch in 1904–1905 in the district of Cooktown, Queensland.

ornamented with non-wooden materials such as textiles and feathers. The motifs themselves are highly variable, ranging from abstract lines, crosses, cross-hatching and stippling, to iconic representations of people, objects and landscapes. Regional patterns are not immediately obvious since items that are sourced from the same locations and periods may exhibit strikingly different styles. Moreover many of the objects are inadequately provenanced meaning that stylistic generalisations must be made with caution. However, it is clear that provenanced objects are found within diverse semiotic contexts that have their own localised stylistic conventions expressed, for example, through rock art, body paint, sand-drawings, shield designs, or tree carving.

Functional accounts of actual message stick use are nonetheless relatively consistent from region to region and are thus intrinsic to how the objects are defined. What makes a message stick a message stick is its role as a central material token in a public long-distance communicative event, a fact which distinguishes it from restricted objects. Markings on the wood reinforce the legitimacy of the communication and typically correspond to individual elements of an accompanying spoken message (Fig 2).

According to the earliest colonial accounts [3–5], an Indigenous person camped in a particular locale might decide to send a message to another person in a neighbouring territory. The sender appoints an appropriate messenger and begins carving a message stick in their presence while explaining the content of the message. The messenger memorises the message and carries the message stick overland to its intended recipient. On the journey the message stick is often prominently displayed, for example hanging from the tip of a spear, through a pierced septum or in a headband, to signal permission to enter into foreign territory. It is this necessity that makes message sticks categorically public as opposed to restricted objects. Arriving at the destination, the messenger identifies the recipient and recapitulates, in oral form, the memorised message while referring to the marks. If the message concerns a large meeting involving several groups, the message stick may continue on its journey in the hands of a new messenger. The recipients may produce another message stick in reply, or simply provide an oral response for the original sender. After use, message sticks have been known to be destroyed, filed in sets (see Fig 1), or repurposed as other objects such as hairpins [6] or clapsticks (AMSD IDs: SAM_A_52915, SAM_A_52916).

Non-Indigenous outsiders who observed and documented this process attributed three functions to message sticks: to signal permission for the messenger to enter into a foreign territory, to authenticate the message itself, and to reinforce the memory of the messenger. Almost all rejected the suggestion that message sticks were a form of language-based writing and many were sceptical that their motifs had any conventional meanings whatsoever. Commentaries on message sticks were, however, rarely informed by sustained consultation with Indigenous senders or messengers. The poverty of documentation has meant that the history, pragmatics and typological significance of message sticks is underexplored. For a summary of the collection history of message sticks and the scientific controversies they engendered see [1].

While invaders from the British Isles began making aggressive incursions along the east coast of Australia in the late 18th century, there is no evidence that they noticed message stick use until about 1840 [4], and only from the 1880s did the practice attract wider international attention from both amateur and professional anthropologists. The earliest published description of message stick use appears in Brough Smythe's *Aborigines of Victoria* [3] and more accounts were published soon after by the German anthropologists Adolf Bastian [7, 8] and Rudolf Virchow [9], and the French duo Emile Houzé and Victor Jacques [10]. All were intrigued by the possibility that the marks on these objects might convey accurate information independent of context. It was only later that scholars such as Curr [5, 11], Howitt [4], Roth [12], and Hamlyn Harris [13] appreciated the central role of the messenger and the importance

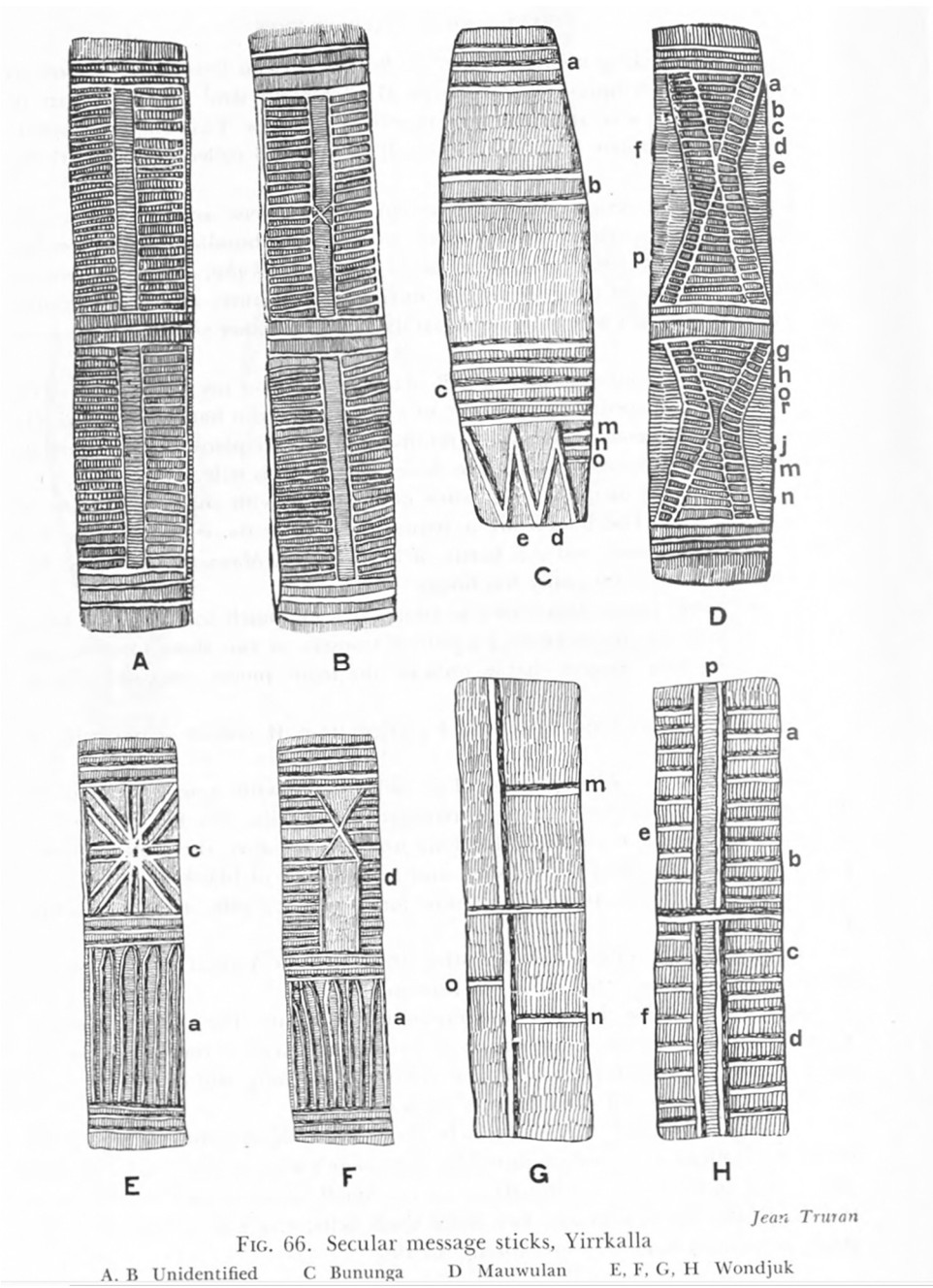

Fig. 66. Secular message sticks, Yirrkalla

A. B Unidentified     C Bununga     D Mauwulan     E, F, G, H Wondjuk

**Fig 2. Annotated sketch of message sticks acquired at Yirrkala by C. P. Mountford as part of the 1948 American–Australian scientific expedition to Arnhem Land [2].** Lowercase letters identify motifs with defined meanings that relate to an oral message.

of an oral channel in explicating a message stick's meaning to an addressee. In the early twentieth century, however, cases were increasingly reported of message sticks sent without a messenger, for example, through the postal service. These types of incidents once again raised the possibility that some message sticks were independently meaningful, even if they were not representing language in a strictly writing-like capacity.

Message sticks bear similarities with other non-linguistic yet rule-governed technologies of communication. In fact the earliest attestations of the term 'message stick' in English are found in accounts of a comparable system in use in Norway where wooden rods (*budstikke*) were sent between communities as a general summons [14]; the practice was also known in Iceland [15], and the Shetland and Orkney islands [16]. It is conceivable that Brough Smyth was referencing this tradition when he first applied the term 'message stick' in his second-hand descriptions of Indigenous communication in Australia, but his primary comparator was khipus, the Andean system of recording mostly numerical information with knotted strings. The American ethnologist Garrick Mallery, meanwhile, compared message sticks to Native American wampum beading, describing how both systems had the functions of lending authority to a spoken message and aiding recall, but that they also expressed conventional meanings that did not rely exclusively on orality [17]. In fact, the Peabody Museum has followed Mallery in applying the term 'message stick' as a cover-all term for both the Australian message sticks and the wampum objects in its collection (also labelled as 'invitation sticks'). After Australian message sticks became known internationally, European scholars retrospectively applied the term 'message stick' to engraved Upper Palaeolithic objects found in cave excavations in Switzerland [9] and France [18]. More recently, the term 'message stick' describes a communication practice in Borneo involving arrangements of wooden sticks and leaves that are marked in purposeful ways [19]. Another very similar practice is the use of *mou khé* among the Hmong, notched pieces of wood that interact with an oral message for important communications [20]. However, despite superficial similarities with these and other techniques, the Australian continent was, for the most part, isolated from the rest of the world until the 18th century and thus the message tradition is highly likely to be an independent innovation. The AMSD stores information about Australian message sticks only.

## The purposes of the database

The purposes of the database (Figs 3 and 4) are: 1) to build a meaningful archive of digital cultural heritage subject to the control and approval of Indigenous community representatives (see 'Access issues and Indigenous Cultural and Intellectual Property' below); 2) to provide a reliable source of comparative data for recovering the history, function and significance of Australian message sticks; 3) to revisit artefacts of unknown origin in order to identify their traditional country of origin and probable communicative intent; 4) to correct contradictions, ambiguities or deficiencies in the labelling of existing artefacts; and 5) to situate Australian Indigenous symbolic culture within a global understanding of graphic communication systems.

The database is intended to serve as a detailed record of all known Australian message sticks in museums and archives worldwide, described with standardised notations to permit comparative and statistical analysis. We cannot claim that the database is complete since we expect new artefacts to come to light in private collections, smaller museums and non-public catalogues. Nonetheless, as far as publicly registered artefacts are concerned, we are confident that it is as comprehensive as possible.

The AMSD follows the precedent of other recent datasets of non-linguistic sign systems of which the Open Khipu Repository (https://github.com/khipulab/open-khipu-repository) represents the most important. Composed of knotted string, khipus have a much more constrained structure than that of message sticks, meaning that a consistent data schema can be applied. Shared characteristics, such as primary cords, pendants, subsidiaries, as well as fibre types, lengths and colours, are captured in tables that serve as 'transcriptions' of individual khipus. The Open Khipu Repository is relational, meaning that each table is linked to other tables

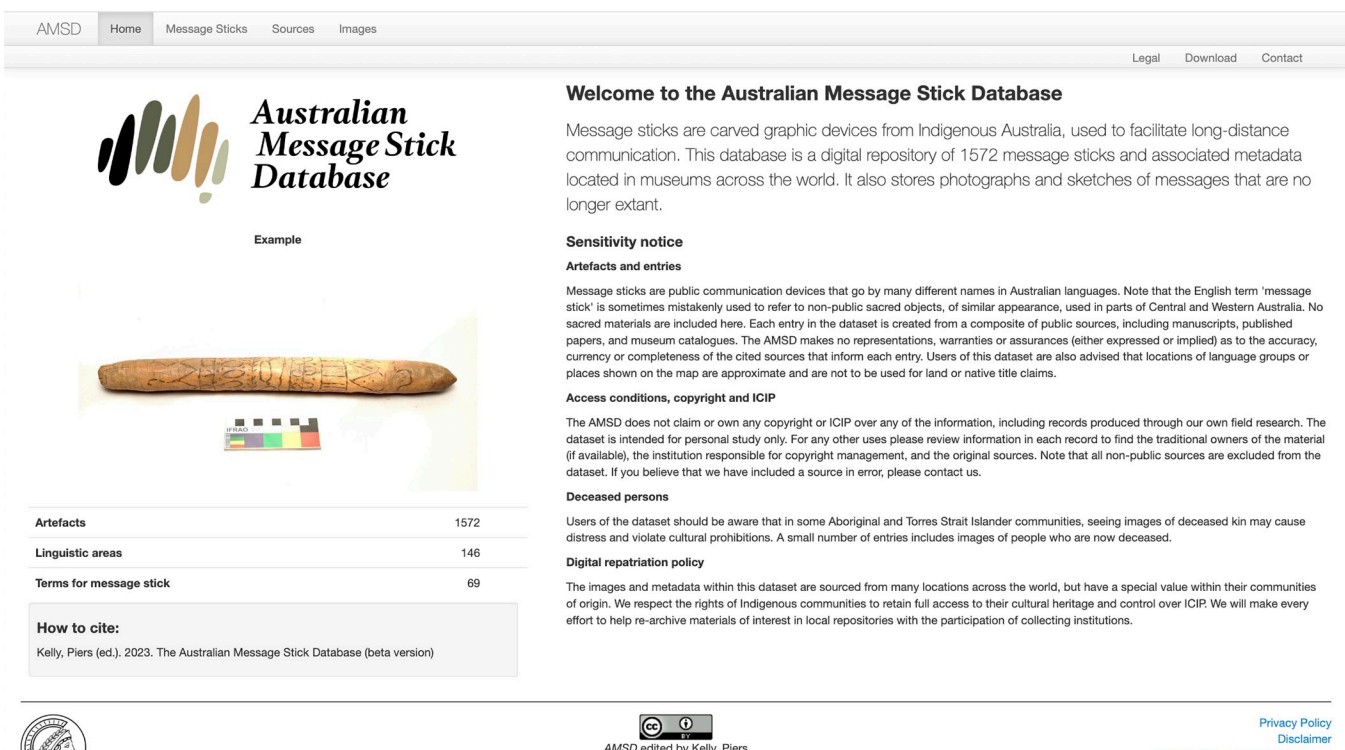

**Fig 3. The Australian Message Stick Database landing page (https://amsd.clld.org/).**

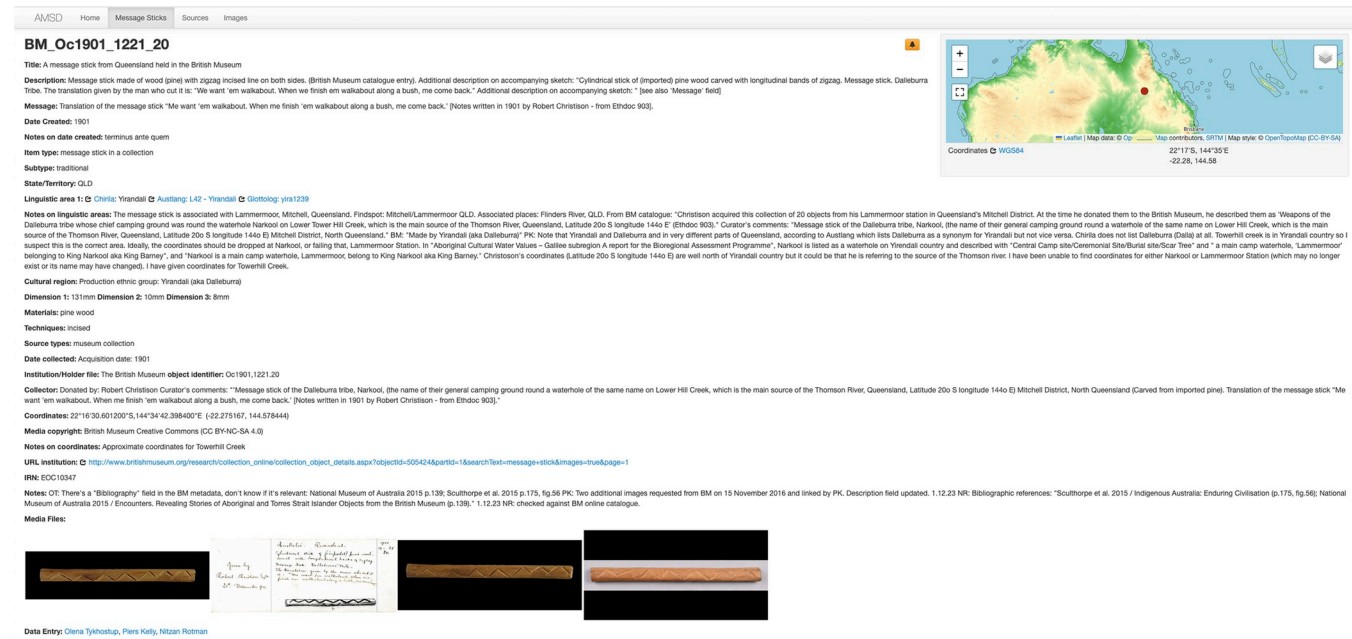

**Fig 4. Detailed entry for a message stick from Queensland in the British Museum (BM_Oc1901_1221_20), at https://amsd.clld.org/contributions/BM_Oc1901_1221_20#4/-22.28/144.58.**

through defined correspondences between data types. Although work on the dataset began in 2002, it has continued to expand ever since with the advent of new archaeological discoveries. Several significant findings have emerged from the analysis of the data, including an impressive concordance between six khipus recovered from the Santa Valley, and a set of Spanish administrative documents dated to 1670 [21].

The AMSD differs from the Open Khipu Repository in a number of ways. Firstly, its descriptions of message sticks are derived from primary sources, and as much contextual detail as possible is retained (see Fig 4). In this respect it has the scope of a detailed museum catalogue or archaeological report. The aim is to represent the sources accurately and to keep this data separate from any analysis, including transcriptions of sign sequences. This distinction stems from the diversity of the materials and the relatively 'free' nature of the sign system, but it is also a function of the kinds of historical questions the database has been designed to probe. Beyond hypotheses that can be addressed solely with statistical or decontextualised data, the AMSD seeks to be a resource for informing more traditional historiography by associating artefacts with full archival sources that are linked to each entry. As such it follows the lead of eHRAF World Cultures (http://ehrafworldcultures.yale.edu/ehrafe/) where the relevant textual evidence is supplied in context and without manipulation.

Indeed, a central tension in the development of the database has been the necessity to represent the sources faithfully while compiling the data with accuracy. This is challenging if the sources themselves are unreliable and require interpretation. Some of the most comprehensively described message sticks exist *only* as textual artefacts, the objects they refer to having been lost or destroyed. Types of unreliable data include, for example, provenance estimations based on the site of collection rather than the site of creation, or speculative comments on the meaning of the message sometimes plagiarised from earlier sources referring to different objects. Various 'notes' fields, discussed below, are therefore embedded in the AMSD to allow researchers to challenge the sources or provide evidence for their reinterpretation, without erasing or manipulating the original text. A similar system for providing metacommentary on the reliability of archival information has been employed in the digital inventory of the Ethnologisches Museum Berlin [22], but we have not yet encountered this within the record management systems for Australian collecting institutions. As a result, errors can easily remain undetected and unaddressed with distressing consequences, including the misattribution of traditional ownership (see, for example, AMSD ID: AMus_E085689). While these issues cannot be comprehensively addressed by the AMSD itself, users can nonetheless provide commentary, contribute new information or report errors through the 'Give feedback' bell icon at the top of each entry (see Fig 4).

## Item types

The AMSD has the strict structure of one discrete item per database entry. But within each entry, a single item may have multiple sources informing it and more than one representation, for example, an official museum photograph, a hand-drawn sketch in a notebook, an illustration in a published article, etc. The core item types are labelled as follows, with counts accurate at the time of publication: *message stick in a collection* (N = 1197), *message stick in a private collection* (N = 70), *message stick from a private sale* (N = 49), *image of a message stick* (*artefact missing)* (N = 170), *footage of a message stick* (N = 6) and *image of a message stick and messenger* (N = 5). Three additional item types are associated with indirect sources evidence for message stick use. These are: *positive text reference* (N = 19) referring to an observation of message stick use in a particular time and place recorded in an archive, *lexical item* (N = 10) meaning an Indigenous term for message stick from an identifiable Australian language, and *message*

*stick accessory* (N = 4) referring to paraphernalia connected to a message stick such as cleft carrying sticks. The final two item types are *negative text reference* (N = 5) recording an archival observation that no message sticks are used by a particular group or in a particular territory at a specific time in history, and *fictional message stick* (N = 22) for imaginative or artistic representations of message sticks that are not known to have existed but which may have a bearing on the cultural history of these objects. Thus, in terms of plotting the distribution of message sticks, the final two item types must be excluded from raw counts, although *negative text reference* may be used to identify historical absences. These item types are designed to capture the range of data types that contribute to the dataset as a whole, but may overlap.

## Fields

In its present form there are 45 data-entry fields for each artefact, representing common categories in museum catalogues such as place of creation, date of collection, name of collector, dimensions of the object, institution name etc. To the largest possible extent, the labels defining these fields have been made to align with the standard metadata scheme specified by the Dublin Core Metadata Initiative. However, since the AMSD aims to represent the evidence on its own terms, non-standard fields have been added to cope with the idiosyncrasies of individual collecting institutions. For example, both the British Museum and the National Museum of Australia display an Internal Record Number (IRN), in addition to an institutional object identifier. Although this is not a standard field it is included in the AMSD in order to help scholars trace the movements of artefacts within as well as between collecting institutions. Due in part to inconsistencies between institutional standards, no single item has information in all available fields, but entries are incomplete for other reasons. Message sticks, among other items of Indigenous material culture, have tended to be under-documented by their original collectors, and nearly half the database is comprised of artefacts that have no information beyond a date of collection, dimensions, and vague location data such as 'Western Australia' or 'North Queensland' (Fig 5). This deficiency makes the task of aggregating them into a database even more urgent: under-documented objects can be interpreted in a much larger context, and new information can be added to entries as it comes to light. Finally, four 'Notes' fields have been added to allow commentaries and annotations on particular kinds of data that are frequently subject to errors or uncertainties. These are: *Notes on date created*, *Notes on linguistic area(s)*, *Notes on coordinates*, and a generic *Notes* field.

## Data sources

The images and metadata for the AMSD are sourced from collecting institutions and individuals, documentary archives, and wordlists. These are expanded below.

### Message sticks from collecting institutions

The bulk of the information is derived from large collecting institutions that have made their data available, either through direct cooperation with the project or through a default policy of public release under varying conditions. The largest numbers have been contributed by the South Australian Museum (290 message sticks), the Australian Museum (238), Museums Victoria (137), the National Museum of Australia (115), the British Museum (77), the Penn Museum (54), the Grassi Museum für Völkerkunde Leipzig (50), the Ethnologisches Museum Berlin (34), the Peabody Museum Harvard (38), and the Pitt Rivers Museum Oxford (28). Across these collections variable terms for the concept 'message stick' present an additional difficulty in the interpretation of museum registers when it comes to deciding which objects should be included. Fig 6 below shows a variety of terms for the same type of object in the

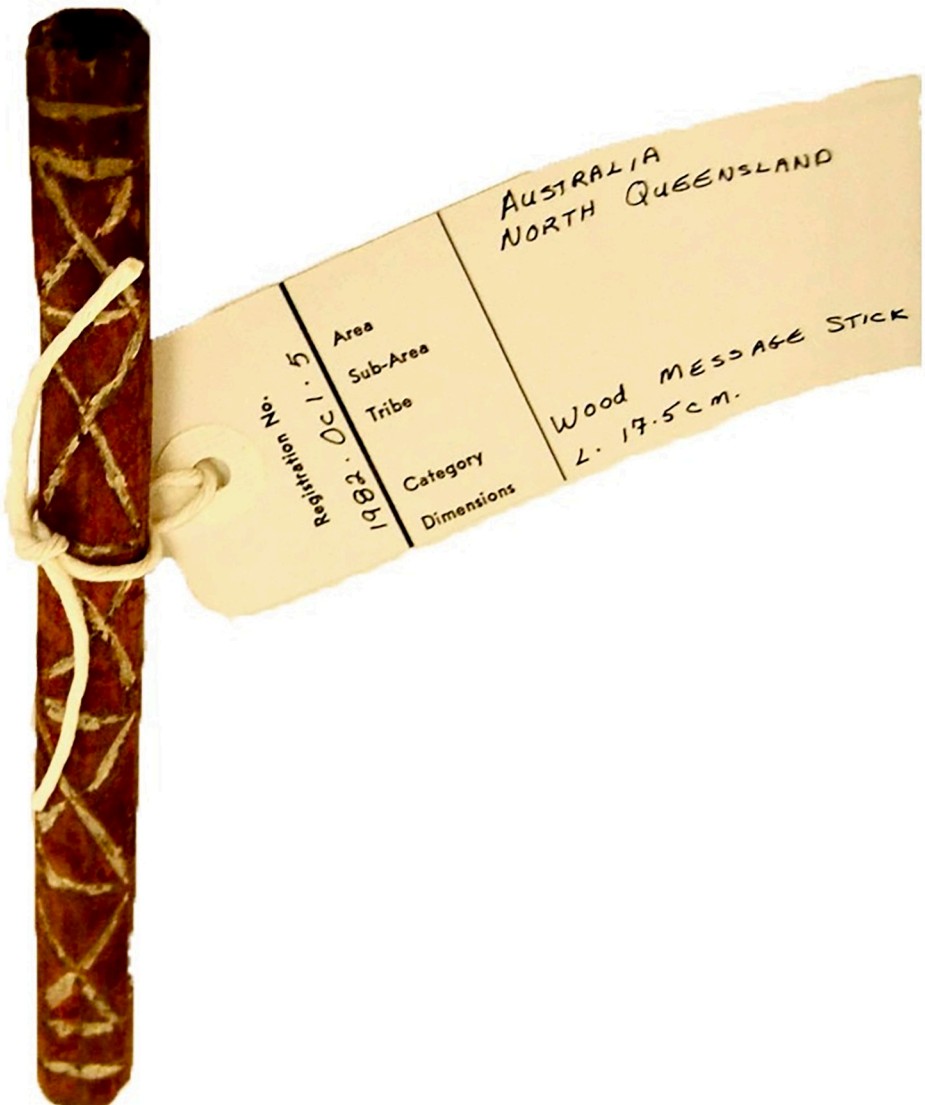

**Fig 5. A message stick (AMSD ID: BM_Oc1982_01_5) made by women who carved the motifs, painted it in red ochre and highlighted the incisions with white ochre.** It was designed for a messenger to carry it through the septum. Unfortunately, no further details of its meaning or origin are available beyond the label "North Queensland". *Image credit*: Trustees of the British Museum.

same museum register: 'Message stick', 'Letter stick', 'Death message stick', 'Death message' and 'Passport' (a common term for 'message stick' in Aboriginal English). In some cases, hairpins, tjurungas (see 'Cultural sensitivity' below) and other morphologically similar objects are misidentified as 'message sticks' and several of these have been found and corrected by the project team. However, it is a shortcoming of the database that other under-documented objects—labelled in museum or documentary sources as 'message sticks' but with limited contextual information—may prove to be false positives if and when more reliable sources come to light. Harder to detect are false negatives: objects in institutions that are message sticks but are mislabelled as other kinds of artefacts. Every effort is made to ensure that the objects are true message sticks before they are included but this cannot be guaranteed. One way that we

**Fig 6. Detail from Peabody Museum register showing synonyms for 'message stick' listed together with morphologically similar items such as 'Ceremonial stick' and 'Hair ornament'.**

have attempted to compensate for such limitations within the database is by coding entries according to descriptive adequacy so that users can filter out less reliable entries (see 'Country-centred design' below).

## Message sticks described in documentary sources

In addition to artefacts held in cultural institutions, the database includes sketches and descriptions published in books, journal articles and newspapers. In total 194 message sticks are reproduced solely from archival sources. Although this represents only around 12 percent of the records in the database, these artefacts are among the best described. Scientific papers and books offer far greater scope for description, explanation and analysis than museum card files or registers. As a result, many of the most important message sticks in the AMSD exist only as sketches or century-old photographs, and the locations of the original artefacts cannot always be recovered. In some cases, museum objects later be associated with more detailed descriptions in external documentary sources. A number of the message sticks in the Ethnologisches Museum Berlin, for example, have been described in published accounts by Adolf Bastian [7, 8], Rudolf Virchow [9], Carl Lumholtz [23] and Enrico Giglioli [24]. The benefit of triangulating sources in this way is that seven 'orphaned' museum objects of unknown meaning or provenance have already been identified by the project on the basis of non-museum sketches, in conjunction with written descriptions (see Figs 7–10 for examples).

## Lexical items

Lexical items denoting the gloss 'message stick' or message-related phenomena constitute a special class of entries in the AMSD. They are included for three reasons. The first is to assist

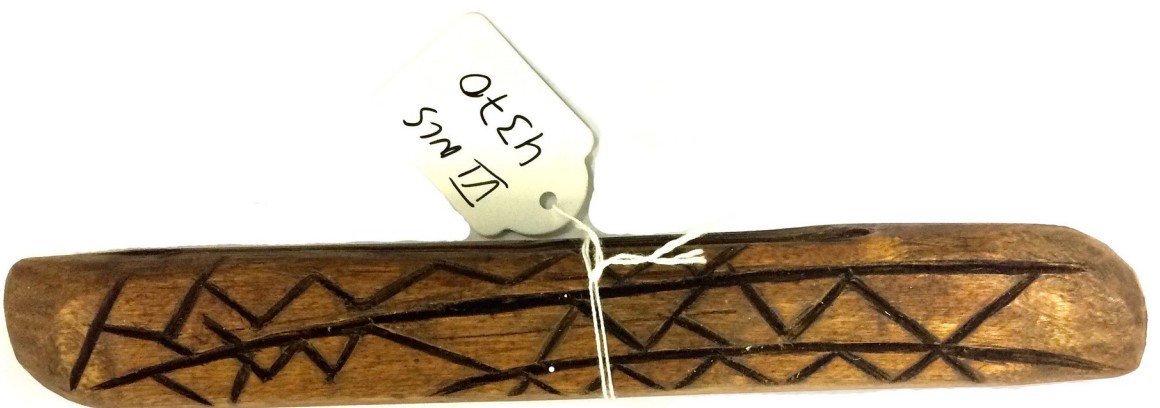

**Fig 7. A message stick discovered unlabelled in the stores of the Ethnologisches Museum Berlin (AMSD ID: EMB_VI_NLS4370; NLS stands for *Nummerlos* or 'numberless').** *Image credit*: Piers Kelly.

in plotting the diffusion of message sticks at the time of contact: we treat the existence of relevant lexemes in a given language as prima facie evidence that message sticks were known to speakers of that language. The second is to maintain a public record of vocabulary for future etymological or reconstructive work. If enough lexical data can be found, over the life of the database, it may allow for lexical reconstruction via comparative-linguistic methods. The third reason is to foreground Indigenous terminology for these objects. Since collectors have sometimes erroneously applied the English term 'message stick' to restricted objects, the use of the appropriate Indigenous term within its cultural-linguistic context encourages semantic precision and mitigates risks of misidentifying objects (see 'Cultural sensitivity' below). At present there are 69 terms denoting the concept 'message stick' in the AMSD.

## Country-centred design

Indigenous scholars have long drawn attention to issues of bias in museum records where standard metadata fields have emerged from colonial taxonomies that overlook, or overwrite, Indigenous epistemologies [22, 27]. As well as perpetuating the ongoing marginalisation of Indigenous voices, such 'standardised' fields introduce distortions into the data itself. Meta-repositories such as the AMSD that aggregate knowledge from more than one source, have the potential to challenge colonial biases as much as perpetuate them. These issues are pertinent to

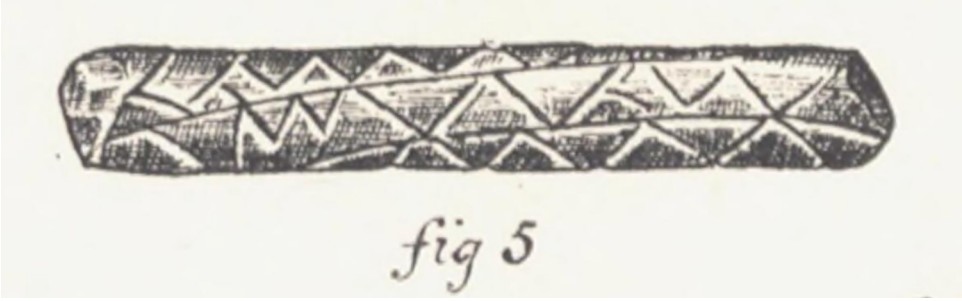

**Fig 8. Sketch of a message stick by A. W. Howitt [4], that identifies Fig 7 above as having been sent between a Mayi-Kulan man to a Mayi-Yapi man in 1883.**

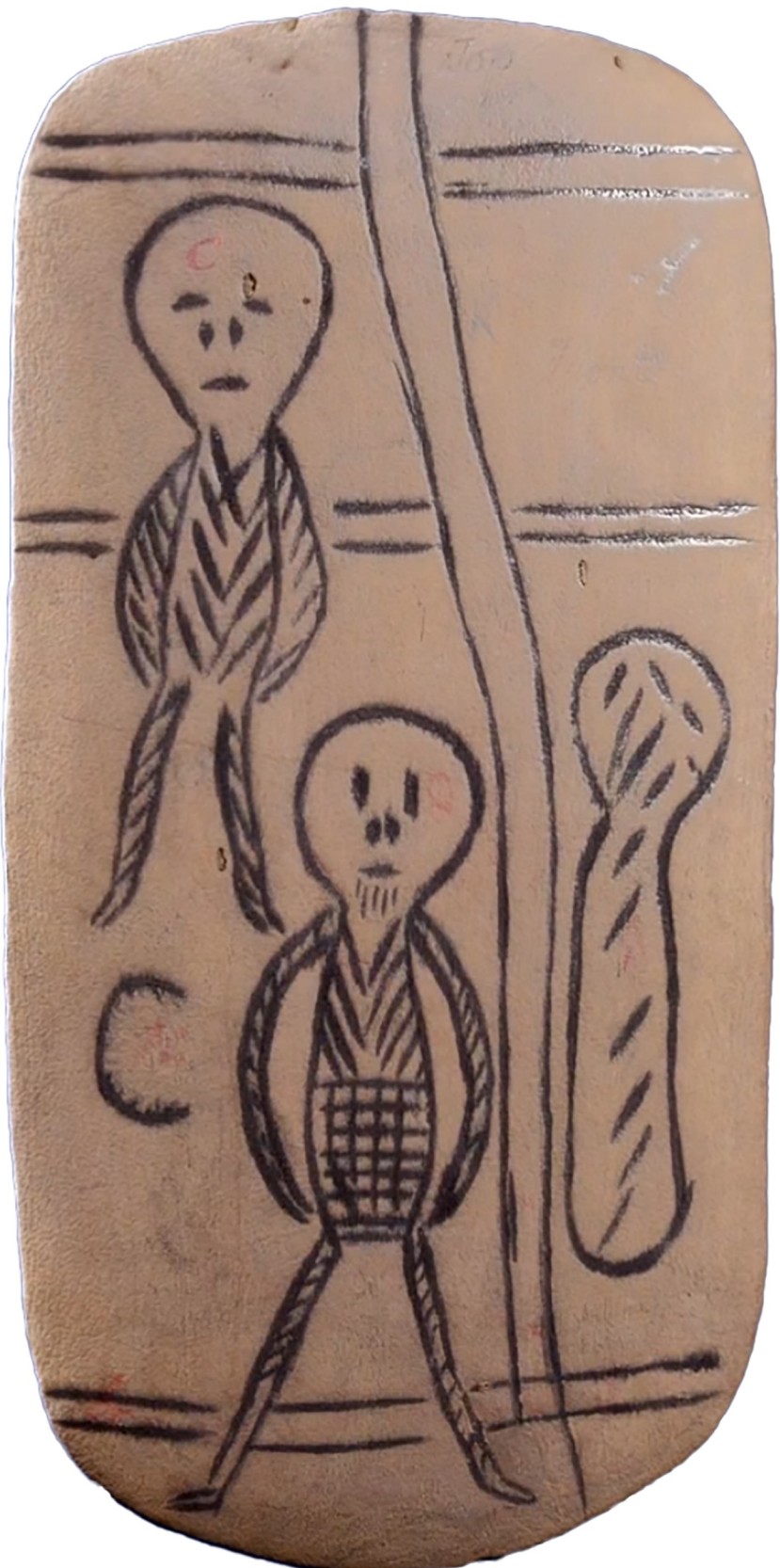

**Fig 9. One of three Muruwari message sticks (AMSD ID: AMus_E032197) in the Australian Museum, previously on display as 'Maker unknown'.**

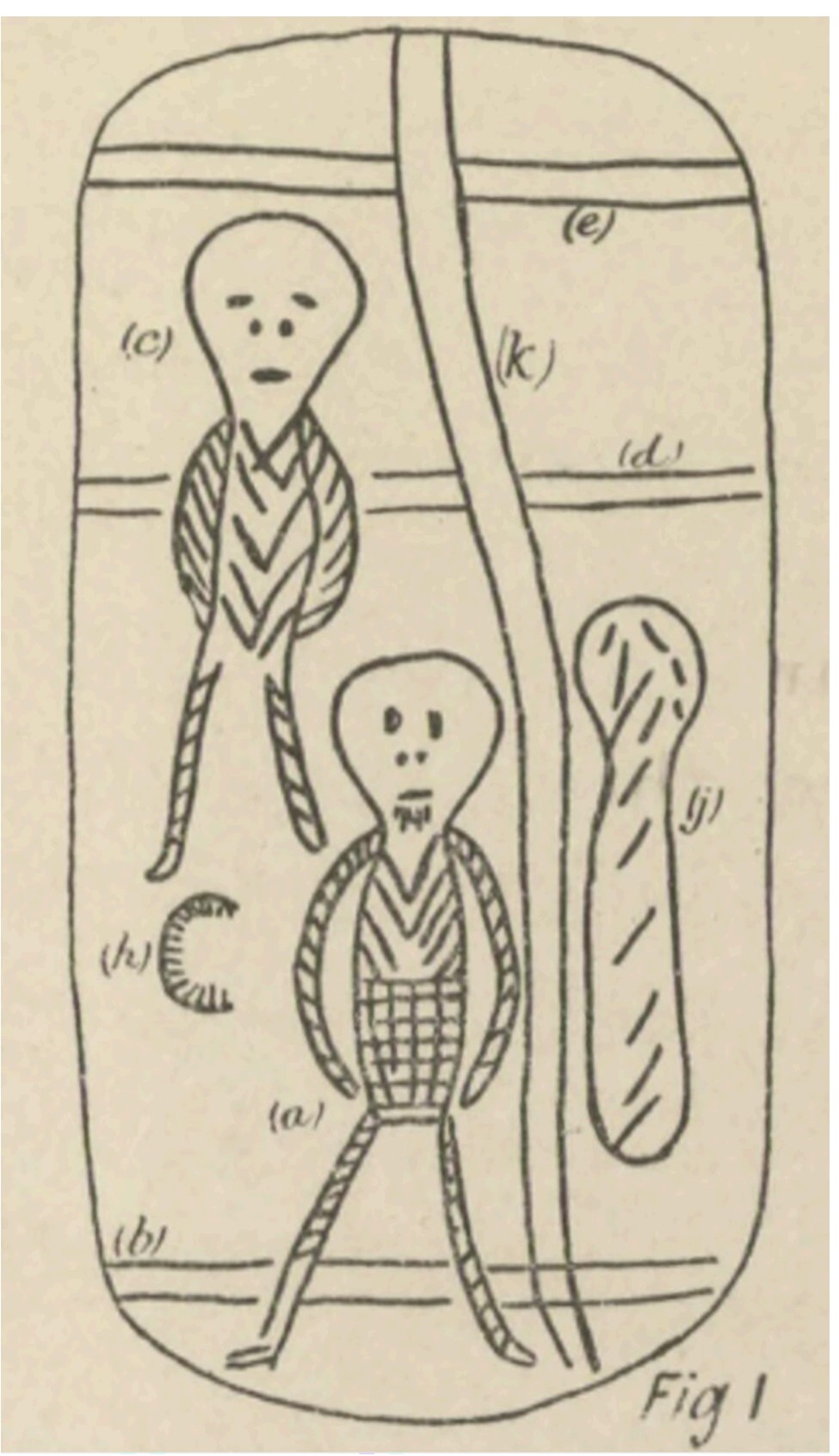

**Fig 10. Sketch in Mathews [25] that led to the identification of Fig 9.** The project team recuperated the name of the creator (Nani) and traced its journey of 385kms westward across Muruwari Country and northward to Kunja Country for a call to ceremony in the early 1890s. This research was later reported in the *First Inventors* documentary series in 2023 [26].

the structure of the AMSD. Indigenous perspectives are rarely present in any of the metadata fields. In only 10 percent (N = 152) of records do we learn the identity of the creator, messenger or recipient of the message stick. Just as astounding is the fact that in only 15 percent (N = 234) of the items did the original collectors elicit any information on their meaning or the circumstances of their use. Instead, forensic attention is often given to material properties such as dimensions, techniques and motif styles as if such information was generative of the object's significance without recourse to an Indigenous consultant.

While this state of affairs is commonplace in colonial descriptions of Indigenous material and symbolic culture, these realities have presented a dilemma in the compilation of the AMSD. Reproducing the original data sources as they stand runs the risk of directly replicating colonial metrics that have either overwritten or misrepresented the original creators of the message sticks. Such an approach would have implications, for example, for naive applications of AI to the AMSD [28]. However, to eliminate or sideline the descriptive information supplied by non-Indigenous collectors would introduce a new kind of distortion, since the database would no longer faithfully reflect its most proximal sources. Further, retaining all such data, however marginal, might allow future scholars to exploit the information in novel ways, for example as clues for locating the original creators or users of the message stick in question, or by extrapolating probable meanings through large-scale comparison.

Our approach to this dilemma has been to restructure the dataset as a whole on the basis of descriptive adequacy. In essence, we allow users to sort the database by foregrounding those records that privilege Indigenous expertise, for example, through an Indigenous authority's direct-speech commentary on the object's purpose and meaning. These higher-value records include entries related to recent fieldwork on message sticks involving living Knowledge Holders in the Top End, as well as historical documentation carried out under the direction of Indigenous messengers who are now deceased. The practice of structurally foregrounding Indigenous knowledge in datasets has been termed 'Country-centred design' by Angie Abdilla and colleagues [29], a counterpoint to human-centred design (HCD). In Aboriginal English term 'Country' embodies a concept in which an area of land or sea is personified and recognised as a source of traditional knowledge (see also [30, 31]). Entries that are 'distant' from Country to the extent that they lack descriptive adequacy in terms of direct Indigenous knowledge can be quarantined from certain kinds of analysis. To this end the data is sorted into four categories based on their level of description. These four categories are (Table 1):

**Table 1.**

| | |
|---|---|
| under-described artefacts | entries that point to the existence of message stick but lack either an image, a location, or a description of the intended message |
| localised artefacts | entries that include a clear image and enough information to associate the item with a place, group or language |
| interpreted artefacts | entries that include a clear image, enough information to associate the item with a place, group or language, and a description of the intended message supplied by a relevant Indigenous consultant |
| glossed artefacts | entries that include a clear image, enough information to associate the item with a place, group or language, a description of the intended message, and an explanation of how motifs relate to independent elements within the message (Eg, "the notches indicate men") supplied by an Indigenous consultant |

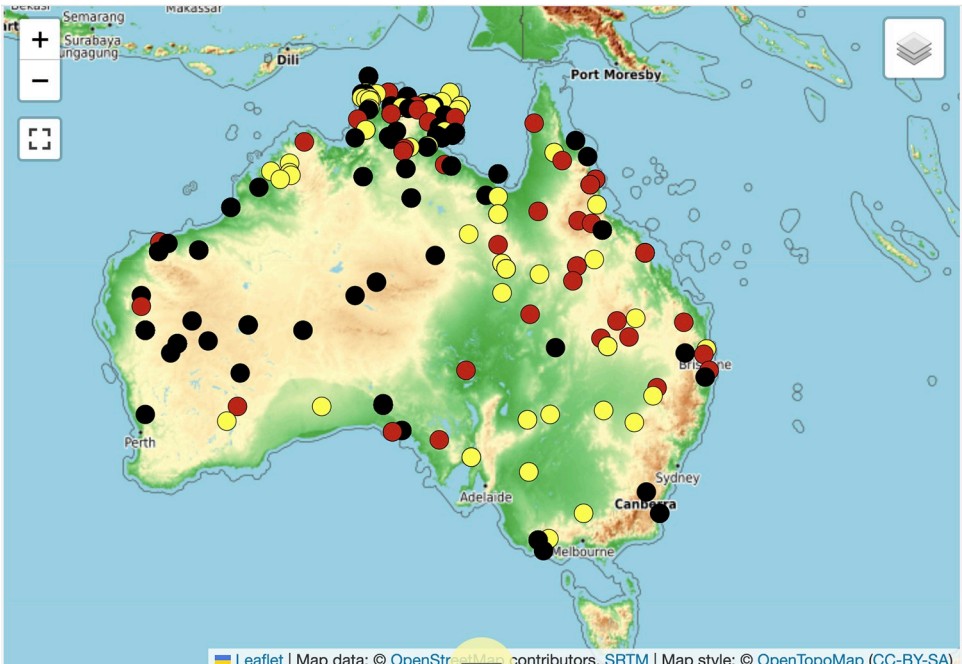

**Fig 11. Sites of origin of message sticks coded according to descriptive adequacy.** Map extracted from https://amsd. clld.org/ on 16 January 2024. All data and layers are released under CC-BY-SA.

At present, these descriptive levels are labelled in the keywords field, allowing for overlaps between categories to be retained. For example all 'glossed artefacts' are also 'interpreted artefacts', but not all 'interpreted artefacts' are 'localised artefacts'. Accordingly, researchers have the option of training their analysis—machine-assisted or otherwise—on the most Country-centred information.

Geo-spatial data visualisation via the map interface, makes these categories easier to work with. Only 56 percent (N = 876) of the entries in the database contain enough information to be able to map them onto their traditional Country. As a result, a priority for database improvement is to find this information through direct and indirect evidence so that the relevant Indigenous communities can be identified and consulted (see 'Research questions and future developments' below). Where location information is available or deducible, entries are colour-coded on the map on the basis of their descriptive adequacy: yellow indicates the location of glossed artefacts, red indicates interpreted artefacts (that are not glossed) and black indicates localised artefacts that are neither glossed nor interpreted (Fig 11 below).

By sifting entries in this way we do not mean to imply that under-described artefacts (shown above as black points) are forever alienated from Country. Rather this process reveals those objects that may demand even greater attention by Elders and Knowledge Holders. Muruwari and Wangkumara scholar Lorina Barker, a contributor to this paper, explains that uninterpreted message sticks are still undergoing their first journey and will find their way home only when they are materially reunited with their Country. Community-led research on the Muruwari message stick (Figs 9 and 10) demonstrated that interpretations through a cultural lens may take the form of intuitive recognition of Country in the object. The same researchers also experienced the information in the archives being "for them", that it had been protected within the institution, and was being communicated by the ancestors at the right time. This experiential mode of interacting with objects and archives points to the importance of

respecting all sources, even if they are minor or partial, in anticipation of later recognition by Traditional Owners.

## Software, web hosting, structure

The AMSD is stored and represented in two locations. All data is first entered into the Online Cultural Collections Analysis and Management System (OCCAMS) hosted by the ANU Centre for Digital Humanities Research in Canberra, Australia. The data is then exported en bloc to the Cross-Linguistic Linked Data (CLLD; https://clld.org/) framework at the Max Planck Institute for Evolutionary Anthropology in Leipzig, Germany. Accordingly, each dataset is mirrored and mutually backed up. These two frameworks are explained below.

### Online Cultural Collections Analysis and Management System (OCCAMS)

The OCCAMS module (https://cdhr.cass.anu.edu.au/research/projects/occams) was developed at the Australian National University by Junran Lei for the purpose of organising and analysing material culture collections. It provides a standard metadata scheme in line with the specifications of the Dublin Core Metadata Initiative. Nonetheless it is flexible enough to allow new fields to be created. OCCAMS is a web-based database tool that aims to address the data management needs of researchers working with and creating cultural collections. It is designed to allow researchers to work on their collections collaboratively in a closed environment and then either publish their data on the web or export it to an archive. Early users of OCCAMS were not simply looking for a Digital Asset Management System, but a research tool with different access levels, and the ability to link and annotate records in order to streamline the processes of working with cultural collections. Users were generating data themselves while also collating data from collections with varying protocols of use and diverse metadata schemas. The strengths of OCCAMS include: the ability to store, organise, attach, annotate and link data in standard formats; to read and write embedded metadata from media files; to batch upload and edit files; to use geo-spatial mapping, timeline, access roles, controlled vocabularies, and flexible metadata schemas mapped to existing standards. The system encourages best practice use of the Dublin Core metadata standards with extensions offered by other universally recognised standards such as IPTC (developed by the International Press Telecommunications Council). This maximises sustainability by ensuring interoperability with other systems. The design of OCCAMS follows the workflow of data acquisition, storage, access, analysis and discovery. The two components of OCCAMS are 1. The Digital File Library for data upload, storage and access; and 2. The Project Workspace for creating records and collections, annotation, analysis and discovery. The system makes a fundamental distinction between digital objects, or files, and collection records that represent objects of different kinds. OCCAMS has already been used by a number of national and international research projects with Indigenous collaborators. These projects include 'The Relational Museum and its Objects', 'Legacy and Impact of the Aboriginal Artists Agency', 'The Legacy of 50 Years Collecting at Milingimbi Mission ', 'The Baguia Collection' and 'Songlines of the Western Desert'.

### Cross-Linguistic Linked Data (CLLD)

The open-source Cross-Linguistic Linked Data (CLLD) framework, a Python package that provides functions for creating and maintaining CLLD web applications, was developed by R. Forkel (https://clld.org/) and is designed for the easy publication of structural data. The CLLD framework integrates language metadata from Glottolog (https://glottolog.org/) for geo-spatial mapping and is flexible for customisable search functions. The CLLD web applications are hosted at Georg-August-Universität Göttingen's Gesellschaft für wissenschaftliche

Datenverarbeitung (GWDG), a computing and IT competence centre for the Max Planck Society. In addition to Glottolog and the AMSD, the CLLD framework currently underpins twenty online datasets. This data will be maintained by the Max Planck Society until 2027 when it will be renegotiated with Georg-August-Universität.

## Cultural sensitivity

Message sticks are public and secular objects created with the intention of being viewed widely. As discussed earlier, their public visibility provided protection to messengers entering foreign territory [3, 4, 32] and for this reason they have been termed 'passports' in Aboriginal English [12, 33–35]. The National Museum of Australia and the South Australian Museum have preserved cleft sticks used for carrying message sticks aloft so that they could be seen from a greater distance (AMSD IDs: NMA1985_0061_0023, SAM_A_51004); similar devices were also used to carry letters written on pen and paper [36] see also AMSD ID: SAM_A_3854). Two message sticks from the Adelaide River at Museums Victoria (AMSD IDs: MusV_X_25838, MusV_X_25839) are attached to beaded headbands [37]. Elsewhere it is reported that message sticks were visibly worn through the septum (Fig 5; also AMSD ID: SAM_A_52913). It is also clear from ethnographic observation that message sticks were presented in the presence of public witnesses [38]. However, if the message conveyed an invitation to male initiation the same messenger might also carry restricted ritual objects carefully hidden in bark or cloth [4, 39, 40]. These additional items were intended as communicative props to be seen only by specific people [41], but they are very much distinct from the message sticks themselves.

The two most general functions of message sticks—as visual tokens of a messenger's permission to travel, and as the authentication of a publicly witnessed communication—are known in all regions of Australia where message sticks have been reliably documented in close consultation Aboriginal knowledge holders, despite differences in the ways such interactions may unfold in practice. Due to their public nature, message sticks are therefore not subject to universal cultural taboos or restrictions. Nonetheless, researchers must remain alert to cultural sensitivities. Collectors such as Daisy Bates, Émile Clement, and Ronald and Catherine Berndt, sometimes mistakenly assigned the term 'message stick' to highly restricted objects used in parts of Central and Western Australia, an error perpetuated by amateur private collectors to this day. These objects, more properly termed tjurungas (or 'churingas' in historical texts) were the sacred property of individual men and in some cases have a similar appearance to message sticks. In certain communities there is potential for great distress to be caused if a woman or uninitiated boy inadvertently sees a tjurunga or an image of one. Although tjurungas have gone through processes of desacralisation in many regions, including their site of origin in Central Australia, they remain sacred in parts of Western Australia [42]. Museums are, for the most part, well aware of these issues. Tjurungas and associated objects are never placed on display and access is typically restricted to initiated men. Auction houses, however, have less scruples than museums, and images of tjurungas are displayed for sale on their sites, sometimes mislabelled as 'message sticks'. With expert guidance from qualified museum staff, we have taken every precaution to ensure that no sacred objects are included in the database.

Within contexts of cultural revitalisation, it is sometimes the case that otherwise profane objects acquire the status of relics. Indigenous revivalists across the world have been known to introduce new kinds of taboos and restrictions around traditional practices, including the possession, production and circulation of items of material culture. Scholars may be inclined to scoff when newly minted taboos are expressed as 'ancient' customs. However the observance of the proscription is no less important for members of communities coming to terms with

their heritage, reasserting Indigenous authority in the face of outside 'experts', or coping with a powerful sense of loss. Examples of this are seen in Native Title reports that document the sacralisation of birthing trees, artefact scatters and scar trees (John Morton pers. comm; see also [43]). As Museums Australia policy document put it, "It is important that museums consider cultural diversity amongst Aboriginal and Torres Strait Islander peoples and realise that the degree to which an object or image is secret or sacred may vary from community to community. It may also change over time. Further, there are instances where materials for which no restrictions apply in their community of origin are subject to restrictions by other groups" [44]. Simply being aware of these concerns does not, however, reveal a straightforward course of action, and if untrained museum staff pre-emptively place a 'sensitive' label on an ambiguous object this can have the effect of damaging the collection by introducing permanent distortions rather than protecting it [45]. Just as much cultural authority is required if a 'restricted' status is attached to object, as an 'unrestricted' status.

In managing the database we remain committed to respecting the wishes of Traditional Owners concerning representations of message sticks and of maintaining absolute consistency in our terminology. One way that we have combatted terminological slippage is through the inclusion of the relevant Indigenous lexeme denoting 'message stick' wherever such information is available. This is provided as an independent field in the database ("Term for 'message stick' (or related) in language:").

## Access issues and Indigenous Cultural and Intellectual Property

Museums have released metadata for our use under a range of differing conditions. The National Museum of Australia has permitted full public access. Other museums have released their metadata subject to the display of full copyright information. Regardless of these varying conditions, by default we have included copyright and licensing information for each individual object from every museum, as well as direct links to their online catalogues. No copyright, including Indigenous Cultural and Intellectual Property (ICIP) is held or claimed by the AMSD itself even for data generated by its compilers. Users are informed, upon entering the database, that the entries are a composite of information amalgamated from various sources but that rights to reproduce individual images remain with the collecting institutions and/or the owners of ICIP. For some collecting institutions we have taken the precaution of watermarking images that have been made available to us, even when we have been granted full rights to store them in the AMSD. This is intended to future-proof the database against changes to museum policies. Nonetheless, the AMSD is committed to digital repatriation of images and all contributing cultural institutions have been informed of our willingness to facilitate exchanges with local Indigenous-managed digital repositories. Such exchanges may help calibrate issues of access, especially with cases of misidentified objects that may later turn out to be subject to cultural restrictions. Local cooperation with the database has already proved productive since members of the project team have been consulted by the Return of Cultural Heritage Team at the Australian Institute of Aboriginal and Torres Strait Islander Studies (AIATSIS) for guidance on the repatriation of significant objects in international collections.

Responding to the demands of Indigenous cultural workers, the Australian Museums and Galleries Association commissioned Indigenous lawyer Terri Janke to produce a review of museum protocols, and this was published in 2018 [46]. The document introduced a legal rights framework that challenged existing access processes. Subsequently, all major Australian museums, with the exception of the National Museum of Australia, have now withdrawn catalogues and registers of Indigenous material from the public, and cross-verifying the data is therefore very difficult. On-site visits are also restricted: researchers must usually demonstrate

support from Traditional Owners of the materials even for catalogue data or provenance investigation. This is often an impossible standard to meet, given that the Traditional Owners cannot be identified without provenance research, nor can materials be requested by them if their existence is not publicly disclosed. An unintended consequence of this approach is that Australian museum staff exercise an effective monopoly on provenance judgements, and a situation of 'provenance paralysis' is reported in some institutions [47]. The limitations on in-person visits also reduce opportunities for communities to examine or photograph unprovenanced (or under-provenanced) objects with the aim of determining whether or not they rightfully belong to their Country. While this state of affairs will no doubt resolve itself when access protocols are revised, the current risk-averse policies means that progress towards a deeper understanding of the materials is stalled [48]. Moreover, Traditional Owners have an additional administrative burden when seeking access to their own cultural property and—in the case of the Australian Message Stick Project—have relied on non-Indigenous intermediaries to navigate the bureaucracy on their behalf. The Right of Reply statement issued by the Indigenous Archives Collective [49] includes an unambiguous directive for the disclosure of holdings but is not yet endorsed by any collecting institution (pers. comm. Kirsten Thorpe). These challenges reinforce the importance of aggregating non-sensitive collection data in a public repository. While collecting institutions in Australia have remained the least accessible to First Nations applicants in our experience, similar issues are reported in international collections [22].

Recently, scientists and Indigenous cultural workers have looked for ways to combine the FAIR (Findable Accessible Interoperable Resuable) and CARE (Collective benefit, Authority to Control, Responsibility, Ethics) principles in order to promote a more ethically sustainable and scientifically rigorous management of datasets [50]. While the work is ongoing, the AMSD has sought to adhere to these general principles by making its (non-restricted) data available in a standardised way and identifying Indigenous data owners in the entries, wherever possible. A related response has been the development of the Mura Maarni framework by the Taragara Research Group, a research partner of the AMSD [51]. The model emphasises the slow development of community relationships and reciprocal obligations with collecting institutions, and stresses the inclusion of cultural metadata with curated objects, respect for data sovereignty, and the minimisation of administrative barriers for community access. It is this devolved, open and adaptive framework that continues to inform our interactions between community representative organisations and the database. Since each community brings its own concerns, values and interests to the data, there is no one-size-fits-all policy when it comes to representing cultural information.

## Research questions and future developments (AMSD 2.0)

The AMSD is designed to facilitate the analysis of message sticks within their original cultural, geographic and semiotic contexts. In its current state, it is a qualitative medium-scale database and is not intended to replicate the kinds of visualisations and projections that are possible with other CLLD databases. At its most basic level it provides an interface for visualising the areal distribution of message sticks according to different search parameters. The resulting distributions can then be superimposed on distributions generated by other datasets. It is outside the scope of this paper to reproduce such analyses here. However, in theory any number of productive investigations could be performed. For example, the distribution of message sticks that convey invitations to certain kinds of ceremonies could be compared to distributions of kinship systems derived from the AustKin database [52, 53] to detect whether ceremonial relationships coincide with or diverge from kinship networks. Relative distances between senders

and recipients could be calculated in order to learn how far messages actually travelled and this could be contextualised with environmental data.

More traditional historians can use the AMSD to discover relevant information triangulated from multiple sources. However, message sticks are also historical documents in their own right, offering a largely unmediated Indigenous perspective on events of local importance. Just as the Open Khipu Repository has contributed important non-colonial source materials to the study of the Inka empire, the AMSD offers opportunities for ethnographic historians to draw on untapped forms of evidence.

The AMSD has been designed to represent source data faithfully but it does so within standardised fields to allow for consistent comparisons. Future developments to the database will include a separate layer of motif coding, whereby message sticks will be 'transcribed' by trained coders whose inter-rater reliability is assessed. Since this layer represents data analysis rather than raw data it is not included in the current instantiation of the AMSD, however in subsequent versions, the full transcription of all 1572 artefacts may permit new inferences to be made about motif distribution. Furthermore, any consistent patterns of association between specific motifs and known meanings may offer the possibility for (probabilistic) 'decipherments' of under-described message sticks. First Nations partners on the project team, meanwhile, are looking towards a second phase of research that goes beyond the existing dataset towards building and discovering richer *cultural* knowledge about individual items. Such research will inform collecting institutions in decisions about display, access and repatriation.

## Conclusion

The Australian Message Stick Database is a new resource that stores standardised information on message sticks that are either held in contemporary collections, or that are known from historical records. To the extent that sources allow, the database displays visual images of the objects as well as information about their origin and meaning. Uniquely, the database is informed by the Indigenous Australian concept of Country and the principles of Country-Centred Design [29]. Moreover, this structure is integrated into OCCAMS and CLLD frameworks, software that represent current best-practice for digital cultural database management. Message sticks are poorly understood objects, due to the ravages of colonial expansion, the under-documentation of artefacts, and consistent misrepresentations of Indigenous knowledge. However, by exploring message sticks as a larger set we believe it will be possible to make meaningful progress in understanding traditional Australian information technology at scale. The database has been developed at a time when Australian collecting institutions are suppressing information about individual items of Australian material culture in their care, making it harder for Traditional Owners to access cultural property, or for researchers to investigate provenance and meanings. As such, we expect that the database will serve as growing and changing resource that will help Traditional Owners to identify and reconnect with ancestral knowledge.

## Acknowledgments

We acknowledge the individual creators of the message sticks included in the Australian Message Stick Database. In almost all cases their names are not recorded. Named creators in the database are as follows with recording spellings preserved; the Country or clan associated with their message sticks is included in parentheses:

Abadjera (Anindilyakwa), Alwyn Doolan (Gooreng Gooreng and Wakka Wakka), Andrew Galitju (Dhuwal), Andy (Mutumui), B. Burruwal (Kune), Barney Two (Tiwi), Belay (Muruwari), Berak (Woiwurrung), Billy (Gunggari), Billy Brookes (Djirbal), Bindjalbuma (Djapu),

Bob (Wunambal), Bununga (Dhangu), Clive Yunkaporta (Wik Ngatharr), Damon Miri Anderson (Gamilaroi, Kullilli, Bundjalung and Wakka Wakka), Dick and Linday Roughsey (Lardil), Fred Jarrarr (Lardil), Harry Carpenter Manilukini (Tiwi), Harry Mahkarolla (Yolngu), Jack Gerambay (Djirbal), Jakie Unung (Burarra), Jimmy Njiminjuma (Kuninjku), July (Tiwi), Kaawirn Kuunawarn (Giraiwurung), Kainaraui (Mengerr), Kunganooay (Muruwari), Madi (Warndarang), Mary (Yirandali), Matarman (Dhangu), Mauwulan (Dhangu), Merai (Warndarang), Mickey (Yirandali), Micky (Yan-nhangu), Miljirina (Mengerr), Mungarui (Dhangu), Munjena (Wirangu), Nanee (Muruwari), Natjiyalma, Ngarkaya and Maaw (Yolngu), Ngaradjin (Dhangu), Nowwanjung (Warrgamay), Nundjan Djiridjarkan (Noongar), Old Peter (Gureng Gureng), Paddy (Djirbal), Sandy (Warluwarra), Spider (Larrakia), Taballah (Muruwari), Tuprukamiri (Tiwi), Un-Tu-Looie (Tiwi), Walter Coulthard (Adnyamathanha), Wondjuk (Dhangu), Wonggu Mununggurr (Djapu), and Wyma (Yirandali).

The following Indigenous knowledge holders contributed crucial knowledge that has informed the inclusion of variables and approaches to data description and interpretation: Lena Yarinkura, her late husband B. Burruwal (name represented with permission), Stanley Rankin, Kevin Djimarr, Bede Tungutalum, Lorina Barker, Roy Barker, Gwen Barker, Paul Gordon and Rick Elwood.

The following experts (Indigenous and non-Indigenous) provided archival information, visual data, or expert advice: Alice Beale (South Australian Museum, Adelaide), Michael Brogan (Taragara Research, Armidale), John Carty (South Australian Museum, Adelaide), Dorothea Deterts (Ethnologisches Museum, Staatliche Museen zu Berlin), Robert Dooley (Museums Victoria), Matthias Hofman (Museum der Weltkulturen, Frankfurt am Main), Philip Jones (South Australian Museum, Adelaide), Eliza Kent (Taragara Research, Armidale), Cynthia Mackey (Peabody Museum of Archaeology and Ethnology), Rita Metzenrath (AIATSIS), David Moore (University of Western Australia), Howard Morphy (The Australian National University), Melanie van Olffen (Australian Museum, Sydney), Eva Raabe (Museum der Weltkulturen, Frankfurt am Main), Birgit Scheps (Grassi Museum für Völkerkunde, Leipzig), Philipp Schorch (Grassi Museum für Völkerkunde, Leipzig), Mariko Smith (Australian Museum), Hilke Thoda-Arora (Übersee Museum, Bremen), Paula Waring (Department of the Senate, Australia), and Renate Wolf (Grassi Museum für Völkerkunde, Leipzig). We thank those who contributed message sticks from private collections: Amber Griffiths-Marsh, Yingiya Mark Guyula (with thanks to Anneke Myers), Guan Lim, David Moore and Katrina Wright.

Olena Tykhostup entered data into the earliest version of the AMSD museums and provided expert guidance on German sources and translations. Julia Bespamyatnykh entered data in a second phase of expansion and translated sources. Together with Rebecca Keim and Piper Young she traced significant artefacts as vectors. Present data managers are Alexandra Roginski and Nitzan Rotman who have paid special attention to the important records at the British Museum, Museums Victoria the National Museum of Australia and the South Australian Museum. Rotman has also assisted with statistical analysis. Finally, Anneke Hamann generously helped to decode historical handwriting in museum registers that had baffled even seasoned archivists.

Ethics approval for the project was granted by The University of New England. We note that ethical considerations that pertain to Indigenous cultural knowledge in Australia can change quickly in response to evolving social and political conditions. Researchers using the Australian Message Stick Database are advised to consult the latest guidance from AIATSIS and local Indigenous communities. Our policies on community consultation and interactions with collecting institutions are set out at https://messagesticks.com.au/.

## Author Contributions

**Conceptualization:** Piers Kelly, Junran Lei, Hans-Jörg Bibiko.

**Data curation:** Piers Kelly, Junran Lei, Hans-Jörg Bibiko.

**Formal analysis:** Piers Kelly.

**Funding acquisition:** Piers Kelly.

**Investigation:** Piers Kelly, Lorina Barker.

**Methodology:** Piers Kelly.

**Project administration:** Piers Kelly.

**Resources:** Piers Kelly, Junran Lei, Hans-Jörg Bibiko.

**Software:** Junran Lei, Hans-Jörg Bibiko.

**Writing – original draft:** Piers Kelly, Junran Lei, Hans-Jörg Bibiko, Lorina Barker.

**Writing – review & editing:** Piers Kelly, Junran Lei, Hans-Jörg Bibiko, Lorina Barker.

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
