## [Decision Letter · Decision Letter 0]

26 Dec 2023

PONE-D-23-30269AMSD: The Australian Message Stick DatabasePLOS ONE

Dear Dr. Kelly,

Thank you for submitting your manuscript to PLOS ONE. After careful consideration, we feel that it has merit but does not fully meet PLOS ONE’s publication criteria as it currently stands. Therefore, we invite you to submit a revised version of the manuscript that addresses the points raised during the review process.  Particularly, all reviewers converge in considering the establishment of this data base should be a positive step in the integration of both modern and traditional knowledge. However, they raise some points to be considered to clarify some statements and generalizations in relation to museums and objects, cumbersome terminology, role of Aboriginal and Torres Strait Islander communities in this research (R.3), and how you plan to address the distressing issue of missing information in museums. 

We look forward to receiving your revised manuscript.

Kind regards,

Marco Peresani

Academic Editor

PLOS ONE

“The lead author (Piers Kelly) receives salary and project funding specifically for the research described in this paper.

He is funded by an ARC Discovery Early Career Researcher Award with the grant number DE220100795.

No other author has received specific funding for this work.”

3. We note that Figure 11 in your submission contain [map/satellite] images which may be copyrighted. All PLOS content is published under the Creative Commons Attribution License (CC BY 4.0), which means that the manuscript, images, and Supporting Information files will be freely available online, and any third party is permitted to access, download, copy, distribute, and use these materials in any way, even commercially, with proper attribution. For these reasons, we cannot publish previously copyrighted maps or satellite images created using proprietary data, such as Google software (Google Maps, Street View, and Earth). For more information, see our copyright guidelines: http://journals.plos.org/plosone/s/licenses-and-copyright.

a. You may seek permission from the original copyright holder of Figure 11 to publish the content specifically under the CC BY 4.0 license. 

Reviewers' comments:

Reviewer's Responses to Questions

**Comments to the Author**

1. Is the manuscript technically sound, and do the data support the conclusions?

Reviewer #1: Yes

Reviewer #2: Yes

Reviewer #3: Yes

2. Has the statistical analysis been performed appropriately and rigorously? 

Reviewer #1: Yes

Reviewer #2: N/A

Reviewer #3: N/A

3. Have the authors made all data underlying the findings in their manuscript fully available?

Reviewer #1: Yes

Reviewer #2: Yes

Reviewer #3: No

4. Is the manuscript presented in an intelligible fashion and written in standard English?

Reviewer #1: Yes

Reviewer #2: Yes

Reviewer #3: Yes

5. Review Comments to the Author

Reviewer #1: The title indicates the manuscript focus is the database and this is well addressed in detail. While comparisons with other databases are useful some detail will be somewhat abstract for the general reader. The paper raises the important conundrum of working with museum objects and the issue of decades and generations of cataloguing discrepancies and disassociation and/or dispersal of related records, while reiterating the importance of going back to primary sources. The value of the database is dependent upon the credibility of the data. The use of figures is important and relevant to the text.

Some statements and generalizations in relation to museums are problematic, such as stating the Australian museums have withdrawn collections from the public; eg. Line 433 begins discussion of current protocols in museums but fails to mention these are imposed/informed practice from the document commissioned by the Australian Museums and Galleries Association from Indigenous lawyer Terri Janke - FIRST PEOPLES: A ROADMAP FOR ENHANCING INDIGENOUS ENGAGEMENT IN MUSEUMS AND GALLERIES - which emerged as a consequence of lobbying by Indigenous arts practitioners in the GLAM sector. While the intention is to give greater say to Indigenous people, it has at the same time created new issues in terms of access without permission of TOs, which is well explained in the discussion - you don't know who to ask if you don't know where it comes from. However the discussion in the manuscript sounds like museums have just decided in 2019 to close down access - more context or explanation is needed.

Terminology in places is cumbersome such as the use of the term 'conserved' in relation to objects in museums. That is one of many responsibilities of museums in relation to objects so suggest using 'held by museums' or a similar term. The other term is 'filed' - eg. in Line 50 the authors state the message sticks were 'filed' by an 'Indigenous recipient' - unsure that best describes that fact they are tied together. And do we know the "Indigenous recipient' actually tied these together? Similarly, in Line 235, the authors discuss 'thicker descriptions' - does this mean more detailed descriptions? Line 266 raises the issue of 'centres of diffusion' which is not addressed anywhere in the manuscript so maybe is best left to explore in more detail in a paper on the actual message sticks; and in Line 278 the absence of 'overtly racist' language in museum records is a puzzling - why mention it if it is absent - again needs more context and beyond the scope of this paper?

However, I question the use in Line 38 of 'oblong' to describe the shape of message sticks. The figure accompanying this shows cylindrical shaped message sticks - definitely not oblong! So are they oblong (essentially rectangular with uneven sides and flat surfaces) and/or cylindrical? Overall, it is important to be clear the authors are giving a general not comprehensive overview of the form and its use as this paper is focused on the database. A minor point is in Line 68 which states that message sticks are 'prominently displayed' yet it is said they can be 'in a net bag' which would mean they would not be seen in that instance.

These comments/observations are raised for the authors to consider in relation to making minor edits to the manuscript.

Reviewer #2: The article ‘AMSD: The Australian Message Stick Database’ by Kelly et al. highlights a critical issue in the study of Australian Aboriginal material culture—ensuring accessible entry for scholars, Aboriginal community members, and Traditional Owners. Opening up this access is vital for fostering collaboration between academics and Aboriginal communities in the study of material culture. Currently, many museums with Aboriginal artifacts have complex and outdated systems in place. The establishment of the AMSD is a positive initial stride towards implementing more effective practices that integrate both modern and traditional knowledge.

I believe the article meets the publication criteria of PLOS ONE. However, I have some comments more focused on the interpretation of the effects of this database, presented as suggestions for future works rather than critiques of the manuscript.

For instance, given that message sticks were in proper use in Arnhem Land until the 1970s, is there anyone among the TOs up there who still possesses traditional knowledge? It would be valuable to include them in the study.

Moreover, the issue of missing information is common and distressing in museums. How do the authors plan to address this beyond interactive comments? Additionally, is there a plan for content moderation that allows input from everyone regarding the items listed in the database?

Some minor comments are presented as follows.

p.2, l.25: typo. Do you mean ‘published and unpublished manuscripts’?

p.3, l.41: It would be helpful to first mention the source of the sample before providing details about the objects. This includes specifying the museum(s), the quantity of objects from each museum, and whether the measurements are from the museum's archive or were taken by the authors. I notice this information is already available below, so I recommend moving these sections here for better organisation.

p.3, l.38-onwards: it might be useful to add more literature references to this section.

p.5, l.74-74: ‘After use, message sticks have been known to be repurposed as other objects’. Can you elaborate on which were these other uses? Just a couple of examples to give an idea of the meaning the message sticks had during as well as after their use.

p.5, l.76: I question the use of the word ‘settlers’ (also above in l.64). Apart for being not a very appreciated word, it is in my opinion too vague. Are them anthropologist, photographers, priests, famers..?

p.11, l.208-onwards: I believe the whole sections ‘Message sticks from collecting institutions’ and ‘Message sticks described in documentary sources’ should appear much earlier in the manuscript (I would suggest just after the historical accounts on message sticks).

p.12, l.217-223: Misidentifying artifacts in museums is a widespread issue. Is the creation of the database, as a project rather than just the database itself, aimed at addressing this problem? Or is it primarily a means of ensuring that the database exclusively includes message sticks? It would be beneficial if the authors shared their thoughts on this matter. Additionally, what about objects that are indeed message sticks but are identified differently in museum archives? Does the project associated with the database intend to tackle this concern too?

p.13, l.238-240: ‘The benefit of triangulating sources in this way is that seven ‘orphaned’ museum objects of unknown meaning or provenance have already been identified by the project on the basis of non-museum sketches and descriptions’. This is very remarkable. Could the authors provide additional insights into the process of triangulation? How did they determine that the museum objects left orphaned were the same as those described in the accounts? While it might be evident when comparing drawings alongside objects, the identification process seems less straightforward when initiated solely from written descriptions.

p.22, l.436-438: I recommend including a brief reflection on the impact of having "impossible standards." Without these standards, in-person visits to the objects could be facilitated, allowing for the capture of new, high-quality pictures. This would prove invaluable for communities to examine 'orphan' items and determine if they rightfully belong to their country. At times, museum photos may not suffice for this purpose, even if the database theoretically makes them accessible to the Traditional Owners.

Reviewer #3: This is a well written and structured article on an excellent project - the development of an Australian Message Stick Database. The authors are to be commended on such an endeavour which has obviously been a long time in the making. My main feedback relates to the current and future role of Aboriginal and Torres Strait Islander communities in this research (including consultation, ethics and data management). My detailed comments comprise:

- There appears to be some inconsistency between the number of artefacts in the database as described in the Abstract (1490), and on the website landing page (1413; https://amsd.clld.org/)

- Introduction: for avoidance of doubt, it would be worth stating up front the definition of ‘message sticks’ in the database, and how they differ from secret or sacred objects with restricted access (eg tjurunga)

- Introduction: in a similar vein, can the authors also clarify (up front) the role of Aboriginal and Torres Strait Islander communities in this study, and whether any ethics processes have been followed (if required)?

- Introduction: it would be worth noting that message sticks are part of a long and varied tradition of engraved and marked wooden items across the continent, including Aboriginal culturally modified trees (eg Spry et al 2023 Investigating Wiradjuri marara (carved trees or dendroglyphs) and dhabuganha (burials), Australian Archaeology journal, amongst other articles)

- Introduction: it could be made a clearer in this section if the authors are arguing that the markings on message sticks have any symbolic meaning or not

- Lines 115-116: “…subject to the control and approval of community representatives”. Do you mean Aboriginal and Torres Strait Islander community representatives? Please clarify if so. It would also be interesting to know more about how this is envisaged to work

- Line 283: it would be worth noting that a historical focus on dimensions, techniques and motif styles, and an absence of Aboriginal people, perspectives and voices, is characteristic of most previous studies of Aboriginal cultural heritage in Australia (eg rock art, Aboriginal culturally modified trees (Etherbridge 1918; Lindsay Black 1941))

- Figure 11: requires a legend

- Line 369-380: can the authors include references to support these statements? Is this view also shared across Aboriginal and Torres Strait Islander communities?

- Line 476-478: can more detail be provided here about this second phase of research? This description currently appears a little vague

- Acknowledgments: the authors may wish to include acknowledgment of all Aboriginal and Torres Strait Islander peoples who created and used message sticks, including those included in the database

- General comment: it would be helpful for the reader to know if this project has, or will, include Aboriginal and Torres Strait Islander community consultation - particularly for well provenanced examples in the database? Noting what the authors state in the section ‘Country-centred design’)

- General comment: can the authors reflect more clearly on how steps could be taken to ensure this database is used in the future for culturally appropriate research?

- General comment: can the authors include discussion of how his study considers FAIR and CARE principles in terms of Indigenous data (https://ardc.edu.au/resource/the-care-principles/)?

- General comment: as this is an international journal, can the author provide more detail on comparable items from other countries? Noting that some comparison is already included

- I was unable to view the entries through either of the two links that accompany this manuscript. I’m unsure if this is because I reviewed the article on a non-PC device with iOS software. The second link asked me for login information, which I do not possess. I encourage the authors to look into this to ensure everyone can view and access all content and website functions through both links. It is for this reason I have stated that data have not been made available for this study

6. PLOS authors have the option to publish the peer review history of their article (what does this mean?). If published, this will include your full peer review and any attached files.

Reviewer #1: No

Reviewer #2: **Yes: **Dr Eva Francesca Martellotta

Reviewer #3: No

---

## [Author Response · Author response to Decision Letter 0]

8 Feb 2024

Please see full response to reviewers in 20240209-Response to reviewers.pdf

---

## [Decision Letter · Decision Letter 1]

16 Feb 2024

AMSD: The Australian Message Stick Database

PONE-D-23-30269R1

Dear Dr. Kelly,

Reviewers have positively commented to manuscript modifications and both agree in considering it is scientifically suitable for publication and will be formally accepted for publication once it meets all outstanding technical requirements.

Kind regards,

Marco Peresani

Academic Editor

PLOS ONE

Additional Editor Comments (optional):

Reviewers' comments:

Reviewer's Responses to Questions

**Comments to the Author**

1. If the authors have adequately addressed your comments raised in a previous round of review and you feel that this manuscript is now acceptable for publication, you may indicate that here to bypass the “Comments to the Author” section, enter your conflict of interest statement in the “Confidential to Editor” section, and submit your "Accept" recommendation.

Reviewer #2: All comments have been addressed

Reviewer #3: All comments have been addressed

2. Is the manuscript technically sound, and do the data support the conclusions?

Reviewer #2: Yes

Reviewer #3: Yes

3. Has the statistical analysis been performed appropriately and rigorously? 

Reviewer #2: N/A

Reviewer #3: N/A

4. Have the authors made all data underlying the findings in their manuscript fully available?

Reviewer #2: Yes

Reviewer #3: Yes

5. Is the manuscript presented in an intelligible fashion and written in standard English?

Reviewer #2: Yes

Reviewer #3: Yes

6. Review Comments to the Author

Reviewer #2: The authors have adequately addressed all of my comments, and I find that the manuscript has significantly improved in this second version. I believe there may have been some miscommunication regarding comment 13, where I simply requested that the First Nation heritage or identification of the authors and/or consultants involved in the study be openly stated. Nonetheless, the authors indirectly responded to my request through adjustments made in response to reviewer 3, which resulted in the addition of a clearer statement regarding the involvement of Muruwari, Wangkumara, and others' cultural expertise. I am pleased to recommend this work for publication without any further revisions.

Reviewer #3: (No Response)

7. PLOS authors have the option to publish the peer review history of their article (what does this mean?). If published, this will include your full peer review and any attached files.

Reviewer #2: **Yes: **Eva Francesca Martellotta

Reviewer #3: No

---

## [Editor Report · Acceptance letter]

13 Mar 2024

PONE-D-23-30269R1 

PLOS ONE

Dear Dr. Kelly, 

I'm pleased to inform you that your manuscript has been deemed suitable for publication in PLOS ONE. Congratulations! Your manuscript is now being handed over to our production team.

Kind regards, 

on behalf of

Dr. Marco Peresani 

Academic Editor

PLOS ONE